# The Magnetic Response of Starphenes [†]

**Mesías Orozco-Ic [1,2,*] and Gabriel Merino [2,*]**

1 Department of Chemistry, Faculty of Science, University of Helsinki, P.O. Box 55, A. I. Virtasen aukio 1, FIN-00014 Helsinki, Finland
2 Departamento de Física Aplicada, Centro de Investigación y de Estudios Avanzados, Unidad Mérida. Km 6 Antigua Carretera a Progreso. Apdo. Postal 73, Cordemex, Mérida 97310, Mexico
* Correspondence: mesiasor@ad.helsinki.fi (M.O.-I.); gmerino@cinvestav.mx (G.M.)
† Dedicated to Professor Riccardo Zanasi on the Occasion of His 70th Birthday.

**Abstract:** The aromaticity of [*n*]starphenes (*n* = 1, 4, 7, 10, 13, 16), as well as starphene-based [19]dendriphene, is addressed by calculating the magnetically induced current density and the induced magnetic field, using the pseudo-π model. When an external magnetic field is applied, these systems create diatropic currents that split into a global peripheral current surrounding the starphene skeleton and several local currents in the acene-based arms, resulting in large shielding cones above the arms. In particular, the arm currents are smaller than their linear analogs, and in general, the strengths of the ring currents tend to weaken as the starphene get larger.

**Keywords:** aromaticity; current density; induced magnetic field; electron delocalization; acene; starphenes; dendriphene

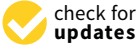



## 1. Introduction

Starphenes are a series of star-shaped polycyclic aromatic hydrocarbons (PAH) consisting of three identical linear acenes interconnected by a six-membered ring (6-MR) belonging to the $D_{3h}$ symmetry [1,2]. As with most large PAHs, scientific interest in two-dimensional organic structures increases due to their potential applications as molecular electronic devices [3]. However, because of the difficulty of isolating large acenes, the synthesis and characterization of starphenes are challenging [1,3]. Recently, Holec et al. reported the synthesis of [16]starphenes [4], which contains three pentacene branches. Similar exciting structures based on starphenes have also been isolated, such as cloverphenes and [19]dendriphene [5,6]. Therefore, the race to obtain larger starphenes is intensely competitive, and the description of their physical properties from a theoretical point of view is still scarce.

Aromaticity in PAHs is a key concept in understanding the electronic nature and bonding of these molecules [7–9]. Since aromaticity is related to the mobility of π-delocalized electrons, the study of the magnetic properties is attractive in starphenes because they may play a key role in their future electronic applications [10,11].

In this work, we address the aromaticity of starphenes comprising arms from naphthalene to pentalene by calculating their magnetic response in the presence of an external magnetic field (see Figure 1). The magnetic response is studied by calculations of the magnetically induced current density [12–16] and the induced magnetic field. Given that the π-electron cloud is responsible for electron delocalization in these organic molecules, and the computational cost of computing these fields is expensive (especially when considering the separation of orbital contributions), we employ the pseudo-π model [17], which allows a precise and computationally cheap estimation of the π-component of the magnetic response in planar and non-planar organic and all-carbon structures [10,17,18]. Our results reveal that starphenes are prone to produce fully diatropic π- currents. Such currents split into a peripheral current and several local circulations flowing primarily in the starphene arms, giving large shielding cones above (and below) the arms. However, the central ring does not support any ring current. When exploring the ring-current strengths of the

different starphenes and comparing them with those of the linear acenes, it is concluded that the aromatic character decreases as the starphene structure increases.

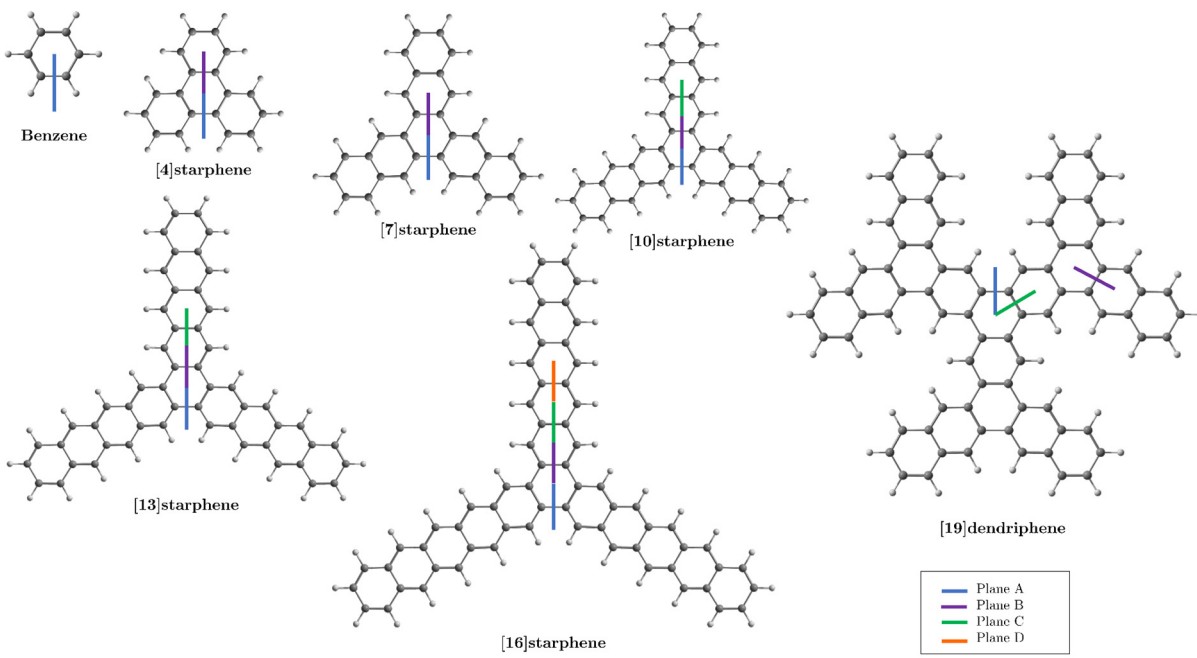

**Figure 1.** Starphenes studied and the integration planes intersecting their different C–C bonds. The planes of integration extend 8 bohr above and below the molecular plane.

## 2. Computational Details

All geometries were fully optimized at the CAM-B3LYP-D3(BJ)/def2-TZVP level (Supplementary Materials) [19–21]. Since the magnetic properties computed with hybrid functionals are in good agreement with those calculated at the CCSD(T) and MP2 level [22,23], nuclear magnetic resonance (NMR) computations were carried out at CAM-B3LYP/def2-SVP level [19] using gauge-including atomic orbitals (GIAOs) [24,25]. All these computations were performed using Gaussian16. [26] The magnetic response was addressed by computing the magnetically induced current density ($\mathbf{J}^{ind}$) [12–14], and the induced magnetic field ($\mathbf{B}^{ind}$) [27–29].

The analysis of the orbital contributions to the magnetic response is particularly useful in understanding the role of each block of electrons on electron delocalization. However, it is computationally demanding. It is possible for organic and all-carbon molecules to accurately approximate the magnetic response of the π-electrons via the pseudo-π model [10,17,18]. This strategy results from calculating the magnetic response of an optimized organic system by removing the hydrogens (if any), and the carbon centers are replaced by a set of hydrogen atoms in the same positions. Therefore, we compute the pseudo-π modeled induced current density ($^{P\pi}\mathbf{J}^{ind}$) and the induced magnetic field ($^{P\pi}\mathbf{B}^{ind}$). The calculations of $^{P\pi}\mathbf{J}^{ind}$ and $^{P\pi}\mathbf{B}^{ind}$ were performed with the GIMIC [12–14] and Aromagnetic [30] programs, respectively. The external magnetic field was applied parallel to the highest-symmetry molecular axis, which coincides with the $z$-axis. In this orientation, the $z$-component is the most significant contribution to $^{P\pi}\mathbf{B}^{ind}$ ($^{P\pi}B^{ind}_z$). Thus, the $^{P\pi}\mathbf{B}^{ind}$ analysis can focus on the $^{P\pi}B^{ind}_z$ component. For a unit external field and planar molecules (as in this work), $B^{ind}_z$ is equivalent to the $zz$-component of the nucleus-independent chemical shift (NICS$_{zz}$) index [31–33]. GIMIC also computes the ring-current strengths ($J^{ind}$) by integrating the current density flowing through a plane intersecting one or more chemical bonds. The ring-current strength number is commonly used to quantify aromaticity. In addition, the current–density flux in different parts of the plane can be determined by calculating the derivative of $J^{ind}$ ($dJ^{ind}/dr$) with respect to one of the plane

coordinates (*r*) [14]. The plane of integration extends 8 bohr above and below from the molecular plane. Also, the coordinate *r* can be chosen such that it starts with *r* = 0 on the horizontal axis at the center of the central ring of starphene, intersects all the C–C bonds, and ends where $^{P\pi}\mathbf{J}^{ind}$ vanishes (ca. 8 bohr away from the last C–C bond). This allows us to screen smooth changes in the ring-current strength and define intervals in the plane corresponding to regions where $J^{ind}$ changes sign, where it becomes zero, or identify where the current has different local or global circulations [14]. In starphenes, there are current flows that can classify as global and local circulations. Due to the multipath character of the $^{P\pi}\mathbf{J}^{ind}$ in starphenes, the ring-current circulations are plotted with different colors to distinguish each pathway. To quantify the ring-current strengths of these other flows, the intervals where the field lines cross a given subplane are determined and highlighted using the same color code (Figure 1). The units for the pseudo-π ring-current strengths are in nA/T, while $^{P\pi}B^{ind}{}_z$ is given in ppm. For visualization purposes, the skeleton shown in the figures depicting the pseudo-π magnetic response corresponds to the original carbon, rather than the all-hydrogen system.

## 3. Results and Discussion

Let us begin the discussion, just for comparison, with the magnetic response of benzene. Strictly speaking, it can be considered a [1]starphene. Benzene responds to the external magnetic field by creating a clockwise (diatropic) ring current [12]. This, in turn, induces a secondary field antiparallel to the external field that generates a shielding cone above (and below) the ring [27]. Both responses are caused mostly by the six delocalized π-electrons. However, noise due to the contribution of the core and σ-electrons may be detectable even at long distances. Therefore, considering only the π-electron contribution to the magnetic response might be better for diagnosing electron delocalization in organic molecules. Given the size of the systems and considering that π-electrons play the most crucial role in the electronic delocalization of organic aromatic systems, we decided to employ the pseudo-π model. In benzene, $^{P\pi}\mathbf{J}^{ind}$ calculations reveal a diatropic ring current (see Figure 2a). Consequently, $^{P\pi}\mathbf{B}^{ind}$ points in the opposite direction to the external magnetic field, resulting in rather intense negative $^{P\pi}B^{ind}{}_z$ values and a long-range shielding cone (Figure 2b).

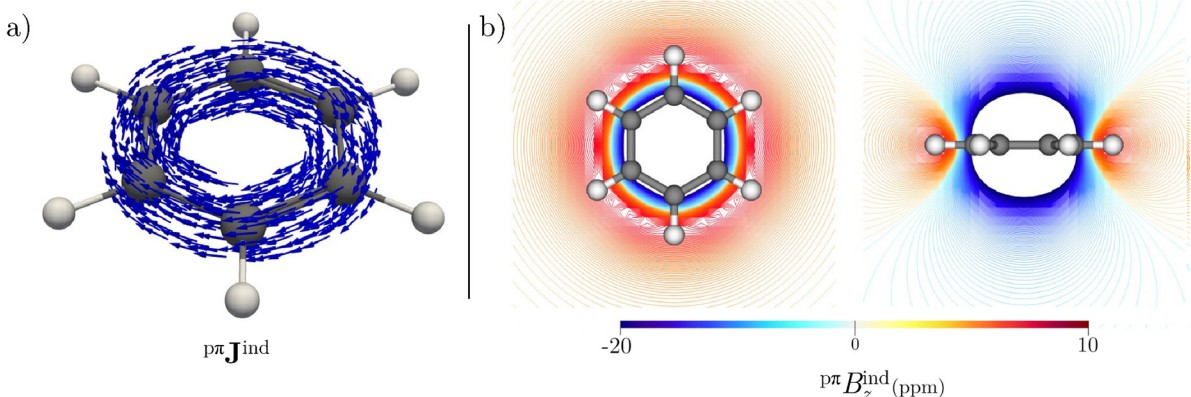

$^{p\pi}\mathbf{J}^{ind}$

-20          0          10

$^{p\pi}B_z^{ind}$(ppm)

**Figure 2.** (**a**) $^{P\pi}\mathbf{J}^{ind}$ vector maps showing the diatropic ring current (in blue) of benzene. (**b**) Isolines of $^{P\pi}B^{ind}{}_z$ calculated in the transverse plane (left) and the molecular plane (right) of benzene. The white areas surrounded by negative (or positive) values correspond to regions where the shielding (or deshielding) magnitude is larger than the values chosen for the color scale.

Previous studies on the π ring-current strength estimate a diatropic behavior of about 12 nA/T for benzene [10,34,35]. In this work, we estimate a value of 12.52 nA/T employing the pseudo-π model. Consequently, the paratropic component of the strength is practically zero. For ring-current strengths caused by all electrons, the paratropic component is nonzero

due to local σ-current loops in the C–C bonds [10,12,35]. The question arises as to how the magnetic response evolves when going from benzene to [4]starphene (triphenylene).

### 3.1. [4]Starphene

The addition of a 6-MR to the three alternate sides of the central benzene-like ring (CBR) gives rise to [4]starphene (or triphenylene). This strongly affects the magnetic response, primarily of the CBR. First, $^{P\pi}\mathbf{J}^{ind}$ shows that the flow lines split into a diatropic peripheral current (global) and local circulations in the outer 6-MRs (see Figure 3a). The integration of $^{P\pi}\mathbf{J}^{ind}$ across a plane intersecting the peripheral current pathway leads to a π ring-current strength of 8.28 nA/T (see Table 1). For the local ring currents, which are also diatropic, the integration of the $^{P\pi}\mathbf{J}^{ind}$ flux at the inner C–C bond provides a strength of 4.12 nA/T. Note that there is no local ring current flow in the CBR. So, from a magnetic response point of view, it cannot be considered an aromatic ring; it is a bridge for peripheral current flow. This explains why the shielding values of the CBR are weaker than those of the outer rings [36,37]. These $^{P\pi}B^{ind}_z$ negative values inside the CBR arise from the combination of the deshielded zones of the outermost rings and the shielding caused by the flux of the peripheral current. The latter ends ruling the magnetic response of the CBR leading to shielded values inside the ring. This phenomenon changes in larger starphenes. Furthermore, $^{P\pi}B^{ind}_z$ shows that the outer rings have a slightly larger shielding cone than benzene due to the local and peripheral fluxes (Figure 3b).

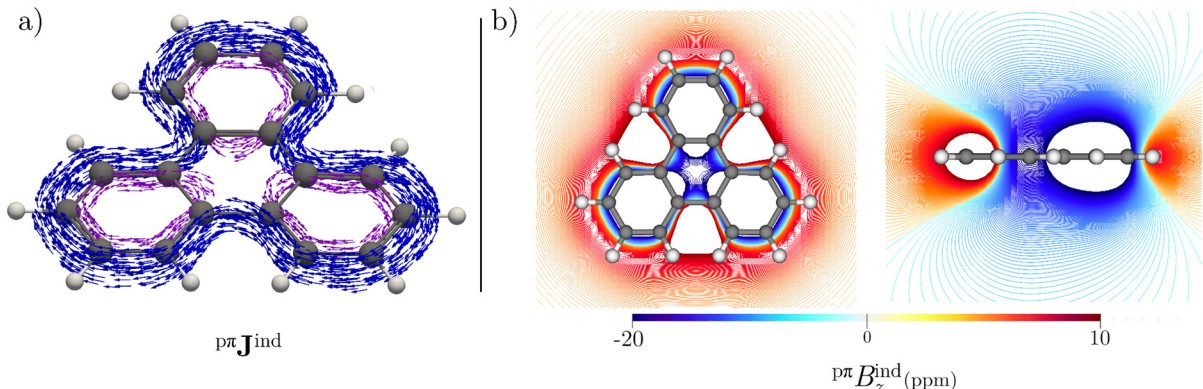

**Figure 3.** (**a**) $^{P\pi}\mathbf{J}^{ind}$ vector maps showing the diatropic peripheral ring current (in blue) and the local diatropic currents (in purple) of [4]starphene. (**b**) Isolines of $^{P\pi}B^{ind}_z$ calculated in the transverse plane (left) and the molecular plane (right) of [4]starphene (scale conventions as in Figure 2).

**Table 1.** Ring-current strengths (in nA/T) of starphenes divided into their diatropic and paratropic components calculated using the pseudo-π model at the CAM-B3LYP/def2-SVP level. The position of the integration planes is shown in Figure 1.

| Molecule | Plane | Diatropic | Paratropic | Net |
|---|---|---|---|---|
| Benzene | A | 12.52 | 0.0 | 12.52 |
| [4]starphene | A (peripheral) | 8.99 | −0.71 | 8.28 |
| | B (local 6-MR) | 4.12 | 0.0 | 4.12 |
| [7]starphene | A (peripheral) | 5.20 | −1.40 | 3.80 |
| | B (naphthalene-like) | 9.41 | 0.0 | 9.41 |
| [10]starphene | A (peripheral) | 3.79 | −1.81 | 1.98 |
| | B (anthracene-like) | 10.46 | 0.0 | 10.46 |
| | C (local 6-MR) | 3.62 | 0.0 | 3.62 |

**Table 1.** *Cont.*

| Molecule | Plane | Diatropic | Paratropic | Net |
|---|---|---|---|---|
| [13]starphene | A (peripheral) | 3.22 | −0.01 | 1.30 |
| | B (tetracene-like) | 10.27 | 0.0 | 10.27 |
| | C (naphthalene-like) | 5.53 | 0.0 | 5.05 |
| [16]starphene | A (peripheral) | 2.98 | −1.87 | 1.11 |
| | B (pentacene-like) | 9.77 | 0.0 | 9.77 |
| | C (anthracene-like) | 5.61 | 0.0 | 5.61 |
| | D (local 6-MR) | 1.73 | 0.0 | 1.73 |
| [19]dendriphene | A (peripheral) | 6.63 | 0.0 | 6.63 |
| | B (naphthalene-like) | 9.09 | 0.0 | 9.09 |
| | C (local 6-MR) | 4.74 | 0.0 | 4.74 |

### 3.2. [7]*Starphene*

The next starphene is [7]starphene. In this system, the $^{\mathrm{P}\pi}\mathbf{J}^{\mathrm{ind}}$ flux lines are branching into peripheral and local naphthalene-like circulations in the arms (see Figure 4a). Also, the CBR maintains the bridging behavior for the weaker peripheral current flow along the starphene. The $^{\mathrm{P}\pi}B^{\mathrm{ind}}_z$ isolines also confirm this (Figure 4b). The shielding cones are mainly above (and below) the naphthalene-like arms. The positive $^{\mathrm{P}\pi}B^{\mathrm{ind}}_z$ values in the CBR are the result of the combination of the deshielding caused by strong naphthalene-type currents and the shielding due to the peripheral current flowing along the starphene. Unlike [4]starphene, the peripheral current of [7]starphene is considerably weaker (3.81 nA/T), which also leads to weaker shielding that is suppressed by the deshielding cones caused by the adjacent naphthalene-like circulations, giving rise to positive values within the CBR. It is well-known that naphthalene produces diatropic currents of 10$\pi$-electron flowing only in the periphery, i.e., there are no individual circulations in each 6-MR [15,16,38]. Reports of $^{\mathrm{P}\pi}\mathbf{J}^{\mathrm{ind}}$ on naphthalene lead to a fully diatropic pseudo-$\pi$ ring-current strength of 13.41 nA/T (at the B3LYP level) [10]. However, the naphthalene-like current in [7]starphene is weaker (9.41 nA/T, see Table 1). So, electron delocalization in the arms of [7]starphene is slightly weaker than that of naphthalene.

### 3.3. [10]*Starphene*

In [10]starphene, it is much clearer that the CBR acts as a bridge to connect the flux of a weak diatropic peripheral current (ca. 1.98 nA/T), while the arms show a well-defined diatropic anthracene-like pathway (see Figure 5). All these currents are diatropic and give rise to large negative $^{\mathrm{P}\pi}B^{\mathrm{ind}}_z$ values inside the rings of the anthracene-like arms. Moreover, traces of an additional current appear in the central 6-MR of the anthracene-like arms because its shielding cone is larger than that of the adjacent rings (Figure 5a).

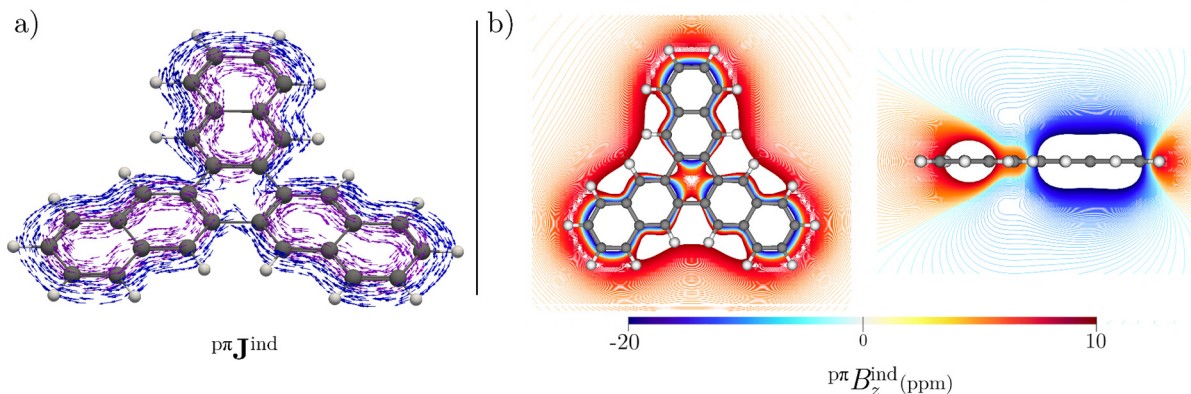

**Figure 4.** (**a**) $^{P\pi}\mathbf{J}^{ind}$ vector maps showing the diatropic peripheral ring current (in blue) and the naphthalene-like ring currents (in purple) of [7]starphene. (**b**) Isolines of $^{P\pi}B^{ind}_z$ calculated in the transverse plane (left) and the molecular plane (right) of [7]starphene (scale conventions as in Figure 2).

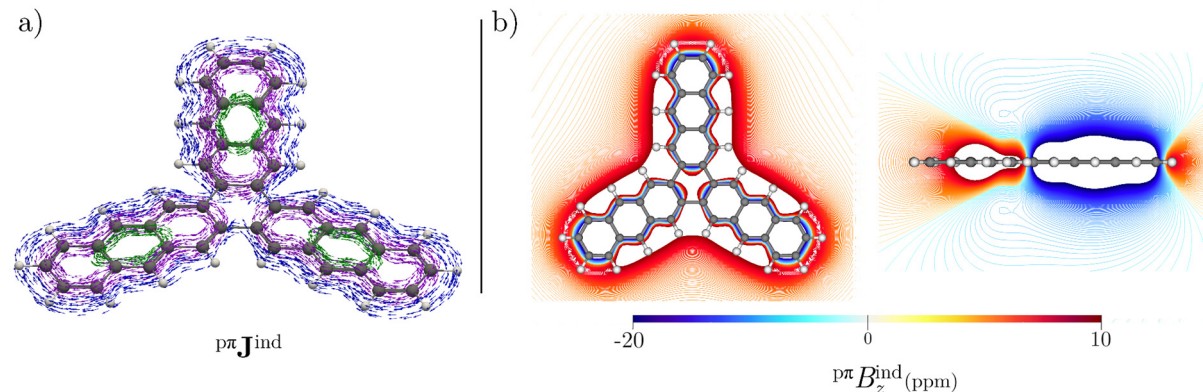

**Figure 5.** (**a**) $^{P\pi}\mathbf{J}^{ind}$ vector maps showing the diatropic peripheral ring current (in blue), anthracene-like ring currents (in purple), and the local circulation in the anthracene-like's middle 6-MR (in green) of [10]starphene. (**b**) Isolines of $^{P\pi}B^{ind}_z$ calculated in the transverse plane (left) and the molecular plane (right) of [10]starphene (scale conventions as in Figure 2).

The concept of local and global aromaticity in anthracene has been widely discussed since the current circulations are explained in terms of 6π-sextets, which does not fit with the magnetic response in this molecule [39–42]. As in naphthalene, the $^{P\pi}\mathbf{J}^{ind}$ shows local and global delocalized pathways in anthracene and larger acenes, with no contamination of the core- and σ-orbitals [10,15–17]. This vector field also reveal that the [10]starphene's arms has a current of 10.46 nA/T, flowing around the anthracene-like perimeters (Figure 5a). In addition, a local circulation of 3.62 nA/T flows in its central 6-MR. This additional local circulation results in a larger shielding cone above (and below) this ring (Figure 5b). This explains why the highest shielded values arise in the center of this ring [8,40]. However, this does not mean that this central 6-MR is more aromatic than the others, but instead there is an overlap of the shieldings from adjacent rings resulting in the fusion of the shielding cones.

Szczepanik et al. [42] showed that the anthracene's patterns can mainly be considered the effect of the superposition of two naphthalene-like 10π-circulations. However, additive schemes on the magnetic response still show discrepancies in perfectly explaining the origin of the central circulations [41,43,44]. The integration of $^{P\pi}\mathbf{J}^{ind}$ across a C–C bond at the perimeter of the central 6-MR leads to a ring-current strength of 15.37 nA/T, which is practically the sum of the peripheral current, the anthracene-like current, and the local circulation in the central 6-MR. To compare, the sum of the local 6-MR and the anthracene-like currents leads to a cumulative net π ring-current strength of 14.08 nA/T, which is also slightly weaker than the net current of the isolated anthracene (~16 nA/T) [10].

### 3.4. [13]Starphene

In [13]starphene, the current pathways are more complex than simple 6π-sextets. In the starphene's arms, there is current flux at the perimeter of the tetracene-like contours (Figure 6a). In addition, local naphthalene-like circulations appear in the two innermost 6-MRs of the tetracene-like arms. Here, it has been pointed out that the tetracene-like aromaticity is the product of individual circuits involving between 10π- and 14π-electrons [42]. No individual ring current flows in the CBR, but a global peripheral current surrounds the entire starphene's structure passing by this ring.

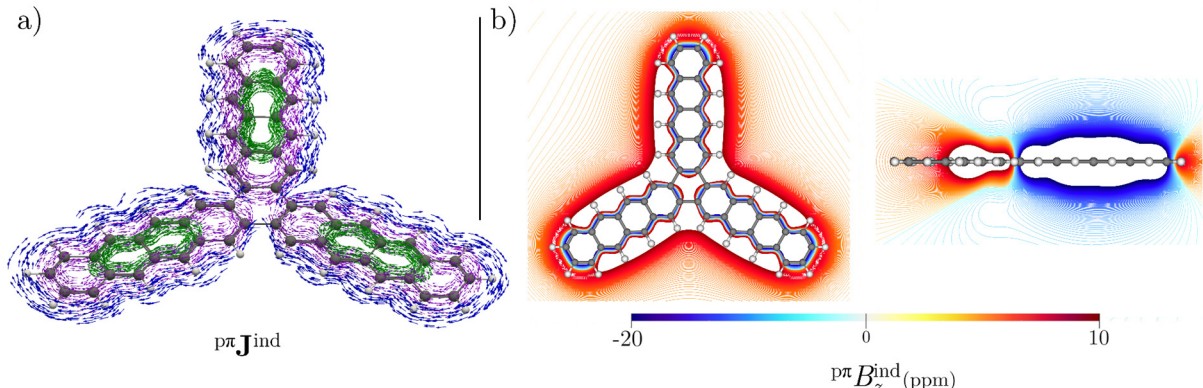

**Figure 6. (a)** $^{p\pi}\mathbf{J}^{ind}$ vector maps showing the diatropic peripheral ring current (in blue), tetracene-like ring currents (in purple), and the local circulation in the naphthalene-like's middle 6-MR (in green) of [13]starphene. **(b)** Isolines of $^{p\pi}B^{ind}_z$ calculated in the transverse plane (left) and the molecular plane (right) of [13]starphene (scale conventions as in Figure 2).

Consequently, $^{p\pi}B^{ind}_z$ calculations show a large shielding cone above these two innermost 6-MRs, while the CBR exhibits large deshielded values in the molecular plane due to the presence of these strong tetracene-like currents and surpassing the weak shielded values caused by the peripheral current (Figure 6b). For systems such as [13]starphene, addressing aromaticity by $B^{ind}_z$ (or NICS) calculations highlights the use of several tools to determine the true origin of shielding or [1]H-NMR signals. The strengths of the local naphthalene-like currents, the local tetracene-like currents, and the peripheral current flowing along [13]starphene are 5.05, 10.27, and 1.30 nA/T, respectively (Table 1). Note that the peripheral current tends to decrease as the size of the starphene skeleton increases, but so do the local currents. For example, while the isolated naphthalene has a pseudo-π current of 13.4 nA/T, [7]- and [13]starphene have a naphthalene-like current of 9.4 and 5.05 nA/T, respectively. Thus, the ring-current strengths decrease as the number of delocalized π-electrons increases in the starphene, also indicating that its aromaticity decreases with respect to their isolated acene's arms.

### 3.5. [16]Starphene

Holec et al. classified [16]starphene as partially conjugated, taking into account that its HOMO-LUMO gap is smaller than that of isolated pentacene and the NICS(0) values [4]. However, NICS cannot provide information on the multi-currents exhibited by [16]starphene. $^{p\pi}B^{ind}_z$ suggests that [16]starphene is highly aromatic due to the formation of long-range shielding cones, of which the central 6-MR pentacene-like arms is the most pronounced (Figure 7). This is explained by the superposition of the shielding caused by several ring currents flowing in the structure. In particular, the $^{p\pi}\mathbf{J}^{ind}$ calculations reveal a weak peripheral current along the starphene, a current running in the perimeter of the pentacene-like arms, an anthracene-like current in the innermost three 6-MRs, and a local current in the central ring of the arms (Figure 7). Like the previous systems, electron delocalization of pentacene-like has been explained in terms of 6π-, 10π-, 14π-, but also 18π-electron circuits [42]. Following the trend, the peripheral current is weaker than that of the previous starphenes (only 1.11 nA/T). The pentacene-like, anthracene-

like, and local currents in the central 6-MR are 9.77, 5.61, and 1.73 nA/T, respectively. A comparison between the strengths of the isolated anthracene and the anthracene-like current of [10]starphene indicates that [16]starphene has a weaker anthracene-like current. Therefore, the degree of aromaticity in starphenes decreases as the arms become larger.

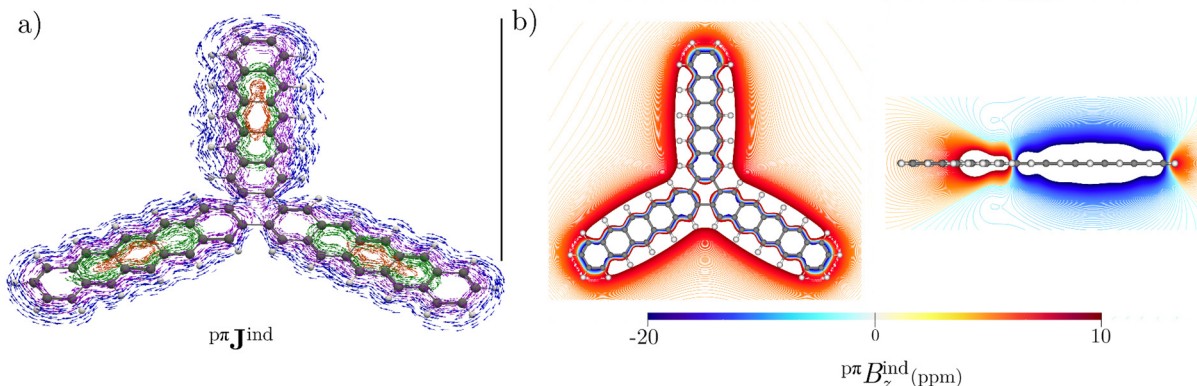

**Figure 7.** (**a**) $^{P\pi}\mathbf{J}^{ind}$ vector maps showing the diatropic peripheral ring current (in blue), pentacene-like ring currents (in purple), the local anthracene-like current (in green), and the local current in the middle 6-MR (in orange) of [16]starphene. (**b**) Isolines of $^{P\pi}B^{ind}_z$ calculated in the transverse plane (left) and the molecular plane (right) of [16]starphene (scale conventions as in Figure 2).

Since the arms are prone to reproducing the magnetic response of linear acenes, open-shell calculations should be considered for larger starphenes. It is well-known that from [6]acene onwards, the determination of the ground state is controversial, as a radical character exists [45–47]. Reports of open-shell NICS(0) calculations on long [*n*]acenes (*n* > 7) indicate an antiaromatic character [46], which could make the synthesis of starphenes with larger acene-like arms even more difficult.

### 3.6. [19]*Dendriphene*

Last but not least, we also addressed the magnetic response of the starphene-based [19]dendriphene [6]. This molecule can be considered an extension of [7]starphene with naphthalene-like arms added to the outer 6-MR or as the fusion of three [7]starphenes with a common CBR. The addition of naphthalene-like arms can even continue iteratively, leading to nanographenes such as [43]dendriphene [6]. The magnetic behavior of [19]dendriphene appears to be a combination of [7]starphene-like circulation and a triphenylene-like circuit in the center. Consequently, $^{P\pi}B^{ind}_z$ shows negative values in the CBR and a shielding cone emerges above that ring. Besides, there are three other 6-MRs in which the outer naphthalene-like arms are attached, exhibiting deshielded values.

However, neither these rings nor the CBR has a locally circulating ring current (see Figure 8). These rings serve for a peripheral strong current flow (of only 6.63 nA/T). The 6-MRs adjacent to the CBR form a triphenylene-like circuit with local currents of 4.74 nA/T, which is slightly larger than those of triphenylene. The outer naphthalene-like arms exhibit diatropic perimetric currents of 9.09 nA/T, considerably weaker than those of [7]starphene and [10]starphene. This confirms that while maintaining magnetic responses typical of aromatic compounds, starphenes and starphene-based structures are less aromatic as they increase in size. [19]dendriphene has been synthesized recently [6]. However, synthesizing larger structures, such as [43]dendriphene, represents a greater challenge, partly due to its size-aromaticity ratio.

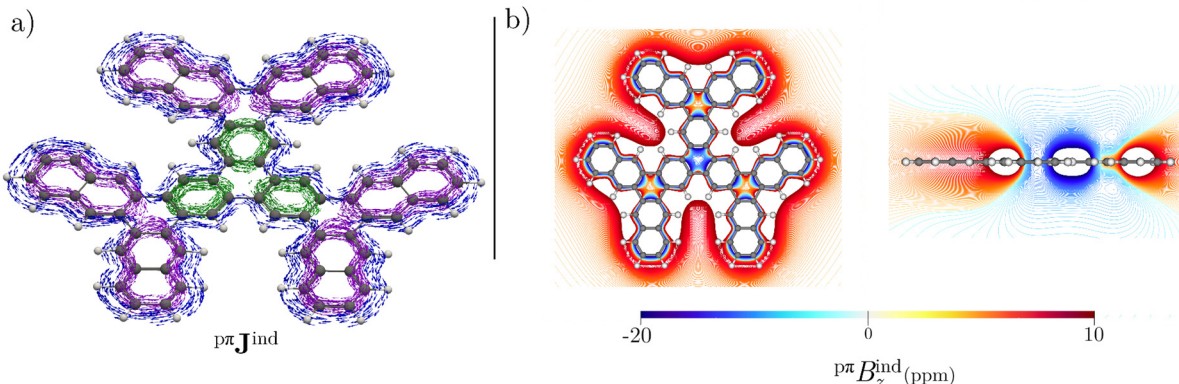

**Figure 8.** (**a**) $^{P\pi}\mathbf{J}^{ind}$ vector maps showing the diatropic peripheral ring current (in blue), naphthalene-like ring currents (in purple), and local 6-MR currents (in green) of [19]dendriphene. (**b**) Isolines of $^{P\pi}B^{ind}_z$ calculated in the transverse plane (left) and the molecular plane (right) of [19]dendriphene (scale conventions as in Figure 2).

## 4. Conclusions

We have analyzed the aromaticity of [*n*]starphenes (*n* = 1, 4, 7, 10, 13, 16), as well as starphene-based [19]dendriphene, by calculating the magnetically induced current density and the induced magnetic field using the pseudo-π model. These molecules respond with fully diatropic π-current densities reflected in large shielding cones on the arms of the star-shaped molecules, conferring an aromatic character to this type of structure. These currents bifurcate into a global peripheral flow and several local circulations in the starphene's arms. The latter shows the same pattern of currents as linear acenes. On the other hand, the central benzene-like ring does not exhibit any ring current. Any shielding obtained for that ring does not indicate the presence of a ring current and therefore cannot be considered an aromatic ring. This differs from other systems with fused rings, such as coronene, wherein the central ring produces a paratropic ring current. In starphene, the CBR only serves as a bridge between the arms for the peripheral current flow. However, the shielding (or deshielding) values exhibited in CBR depend mainly on the strength of the peripheral current passing through it.

Looking in detail at the ring-current strengths, and comparing with the case of linear acenes, it is noteworthy that the currents tend to weaken as the size of the starphene grows. Moreover, the degree of aromaticity also decreases as the starphene gets larger. For example, the largest case, [16]starphene, exhibits a peripheric current, a pentacene-like current, an anthracene-like, and a local current in the central ring of the starphene's arms. Its anthracene-like current is weaker than the strengths of the anthracene-like current of [10]starphene and the linear isolated anthracene. The same can be stated for its peripheral current as for its current located in the central ring.

Something similar occurs in [19]dendriphene, it produces a global peripheral current, naphthalene-like in the arms, and local currents in some 6-MRs. However, compared to other smaller starphenes and to naphthalene-like currents (such as naphthalene itself, [7]starphene, and [13]starphene), the ring-current strengths show a decrease, suggesting a lower degree of aromaticity. This is crucial from both a theoretical and experimental point of view. Although these systems are aromatic, their inverse size-aromaticity relationship may explain why large starphenes are less stable, making their synthesis more difficult.

**Supplementary Materials:** The following are available online at https://www.mdpi.com/article/10.3390/chemistry3040099/s1, the optimized Cartesian coordinates of the molecular structures.

**Author Contributions:** Conceptualization, M.O.-I.; Writing—original draft, G.M. and M.O.-I. All authors have read and agreed to the published version of the manuscript.

**Funding:** This research was funded by the Magnus Ehrnrooth Foundation.

**Institutional Review Board Statement:** Not applicable.

**Informed Consent Statement:** Not applicable.

**Data Availability Statement:** Not applicable.

**Acknowledgments:** M.O.-I. thanks the Magnus Ehrnrooth Foundation for the financial support.

**Conflicts of Interest:** The authors declare no conflict of interest.

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
