# Peer review of "The Magnetic Response of Starphenes"

_chemistry, doi:10.3390/chemistry3040099_

Round 1

Reviewer 1 Report

The manuscript addresses the calculated current density and magnetic field induced by an external magnetic field perpendicular to the molecular plane in a series of beautiful starphene molecules. Due to molecular complexity, calculations were done using the pseudo-pi method, which was shown to be good to model the magnetically induced current density in aromatic hydrocarbons, since it provides right current shapes and bond current susceptibilities in nice agreement with those given by full QM calculations.   

Results show that the central benzene ring loses its characteristic diatropic ring current in all studied molecules, while the adjacent arms retain the induced current of the isolated acenes. However, ring current strengths get weaker in comparison to those of linear acenes, resulting in a decreased degree of aromaticity of the starphenes, which may concur to the difficult synthesis of these class of compounds.

In the opinion of this reviewer, the manuscript presents enough new results and deserves to be published. Only a few minor points listed below should be revised, which do not require to be reviewed again.

One point that is not fully clear to this reviewer is the presence of such large white regions in the induced magnetic field maps. According to the color scale, this should correspond to zero shielding, but this cannot be the case. Is it an artefact due to some cutoff?

In [4]starphene (triphenylene) the induced field within the CBR shows a deshielding zone (blue), then it becomes apparently a shielding zone (red) in [7], [10], [13] and [16] starphenes, then it turns deshielding (blue) again in [19]dendriphene. Is there any clue of possible explanation?

Coordinates of [19]dendriphene should be added to the supplementary material.

Author Response

Comment 1: The manuscript addresses the calculated current density and magnetic field induced by an external magnetic field perpendicular to the molecular plane in a series of beautiful starphene molecules. Due to molecular complexity, calculations were done using the pseudo-pi method, which was shown to be good to model the magnetically induced current density in aromatic hydrocarbons, since it provides right current shapes and bond current susceptibilities in nice agreement with those given by full QM calculations.   

Results show that the central benzene ring loses its characteristic diatropic ring current in all studied molecules, while the adjacent arms retain the induced current of the isolated acenes. However, ring current strengths get weaker in comparison to those of linear acenes, resulting in a decreased degree of aromaticity of the starphenes, which may concur to the difficult synthesis of these class of compounds.

In the opinion of this reviewer, the manuscript presents enough new results and deserves to be published. Only a few minor points listed below should be revised, which do not require to be reviewed again.

Response: We appreciate the reviewer’s positive comments.

Comment 2: One point that is not fully clear to this reviewer is the presence of such large white regions in the induced magnetic field maps. According to the color scale, this should correspond to zero shielding, but this cannot be the case. Is it an artefact due to some cutoff?

Response: Thank you for this comment. The white areas surrounded by negative (or positive) values correspond to regions where the shielding (or deshielding) magnitude is larger than the values chosen for the color scale. This is a scaling problem. However, if another scale is selected, relevant information could be lost in the areas of interest. To avoid misinterpretation, we have added the following in the caption of the figures: "The white areas surrounded by negative (or positive) values correspond to regions where the shielding (or deshielding) magnitude is larger than the values chosen for the color scale".

Comment 3: In [4]starphene (triphenylene) the induced field within the CBR shows a deshielding zone (blue), then it becomes apparently a shielding zone (red) in [7], [10], [13] and [16] starphenes, then it turns deshielding (blue) again in [19]dendriphene. Is there any clue of possible explanation?

Response: The shielded and deshielded regions of the induced magnetic field correspond to negative (blue) and positive (red) values, respectively. In the case of triphenylene, the values are shielded within the CBR. However, for the other starphenes, the situation is the opposite. To explain this, one must consider that the CBRs allow the flow of a global peripheral current flow along the entire starphene. The negative values in the triphenylene CBR are caused by the combination of the deshielding regions of the adjacent 6-MRs and the shielding caused by the peripheral current flow (9 nA/T). For the other larger starphenes, the peripheral current decreases considerably (< 4 nA/T) with respect to triphenylene, which generates weaker shielding. The deshielding regions caused by the arm currents converge in the CBR region giving rise to positive red values. This explanation has now been added to the main text.

In the case of [19]dendriphene, we mentioned that the magnetic response appears to be a combination of circuits similar to [7]starphene and triphenylene. The [19]dendriphene center functions as a triphenylene-like unit. Thus, following the above reasoning, the CBR shows positive values due to the presence of an important peripheral current (ca. 6.63 nA/T). 

Comment 4: Coordinates of [19]dendriphene should be added to the supplementary material.

Response: We have added the [19]dendriphene’s Cartesian coordinates.

Reviewer 2 Report

This paper analyzed and compared the aromaticity of [n]starphenes (n=1,4,7,10,13,16), as well as starphene-based [19]dendriphene, by calculating the magnetically induced current density and induced magnetic field using the pseudo-π model. The computational study has been carried out properly and comprehensively, and the results are interesting. I recommend publication with the following revisions:

  1. The authors wrote in page 2: is equivalent to the zz-component of the nucleus-independent chemical () index. But which position’s ? How to quantify the induced magnetic field? Authors declare that there is other current(s) flow in the peripheral current flow by calculating the magnetically induced current density pπ. Why it cannot be reflected by maps of pπ? I think some explanations are necessary.
  2. There is no local ring current flow in the CBR both for [4].starphene and [7].starphene, but the difference of induced magnetic field between them are so clear, as shown in Figure 3(b) and Figure 4(b), the different color deserve more commentary. Especially in Figure 4(b), the ‘red’ induced magnetic field means the positive pπ, which is opposite with the arm’s induced magnetic field. In my view, it indicates the induced magnetic fields of the three arms are independent. Therefore, the bridging behavior of CBR for the peripheral current flow, as authors declared in page 5, is not faithful.
  3. In table 1, authors showed the ring-current strengths of the diatropic and paratropic components. As an integral whole, the whole ring-current response should be also integrated through a plane intersecting the arm C-C bond. The results are significant and deserve to be shown, because it can give more evidence to make sure whether there is a current flow along CBR’s bridge for big size starphene.
  4. If the isosurfaces maps of pπcan replaced or be added in the figures, they would be much more visual and comprehensible.
  5. There are several papers showing that starting from heptacene these species have a large diradical character and are better represented as a singlet open-shell species obtained using an unrestricted formalism. For this reason, although the systems in this paper are not linear acenes, they are non-linear acenes, they may have the similar properties. I kindly ask the authors:
  6. To acknowledge that the ground state of acenes is a controversial issue.
  7. To cite references of articles showing that acenes have diradical character already starting from n = 7 and that long acenes have polyradical character ( see for instance, M. Bendikov et al. J. Am. Chem. Soc. 126 (2004) 7416; D. Jiang & S. Dai, J. Phys. Chem. A 112 (2008) 332; J. Hachmann et al. J. Chem. Phys. 127 (2007) 134309; J. B. Schriber et al. J. Chem. Theory Comput. 14 (2018) 6295)
  8. To warn the reader that the conclusions one obtains on aromaticity trends considering long acenes as closed-shell species is quite different from the one obtained performing open-shell calculations (see J. Poater et al. J. Phys. Chem. A. 109 (2005) 10629)

Author Response

Comment 1. The authors wrote in page 2: is equivalent to the zz-component of the nucleus-independent chemical () index. But which position’s ? How to quantify the induced magnetic field? Authors declare that there is other current(s) flow in the peripheral current flow by calculating the magnetically induced current density pπ. Why it cannot be reflected by maps of pπ? I think some explanations are necessary.

Response: As mentioned in the computational details section, for a planar molecule the z-component of the induced magnetic field (Bindz) is equal to the NICSzz index when the external field points perpendicular to the molecular plane and the external field is unitary (|Bext|=1T). The purpose of using Bindz is that it corresponds to the most important Cartesian contribution of the Bind vector, reducing the analysis to a scalar field, which can be computed in the molecular surroundings. Thus, negative shielded (positive deshielded) values of Bindz correspond to the regions where Bind points antiparallel (parallel) to the external magnetic field. 

On the other hand, in all starphenes there is a peripheral current flowing around the entire perimeter of the starphene and passing in three of the six CBR’s bonds. We have illustrated in all Jind  maps the peripheral currents in blue. Additionally, several other circulations appear in the arms of the structures. Taking the larger case, [16]starphene (Figure 7), there is a unique peripheral current, a pentacene-like current in each of the arms, a tetracene-like current in the interior of them, and a current in the central 6-MR of those same arms. 

Comment 2. There is no local ring current flow in the CBR both for [4]starphene and [7]starphene, but the difference of induced magnetic field between them are so clear, as shown in Figure 3(b) and Figure 4(b), the different color deserve more commentary. Especially in Figure 4(b), the ‘red’ induced magnetic field means the positive pπ, which is opposite with the arm’s induced magnetic field. In my view, it indicates the induced magnetic fields of the three arms are independent. Therefore, the bridging behavior of CBR for the peripheral current flow, as authors declared in page 5, is not faithful.

Response: In line with the reply to the first reviewer (comment 3), we have added a sentence explaining why in the [4]starphene there are shielded values within the CBR. These values change to deshielded positive numbers in the larger starphenes. The reason is that the strength of the peripheral current produces a shielding region in the CBR, competing with the deshielded outer zones caused by the arm currents. Thus, in the case of [4]starphene and [19]dendriphen, the peripheral current is strong enough to counteract the external deshielding of the adjacent acene-like currents, giving rise to the negative values. However, this is not indicative of an aromatic character of the CBR. For the other starphenes, the peripheral current has a weak strength (< 4 nA/T), and the deshielding governs the magnetic behavior of the CBR, giving rise to positive values in its interior.

Comment 3. In table 1, authors showed the ring-current strengths of the diatropic and paratropic components. As an integral whole, the whole ring-current response should be also integrated through a plane intersecting the arm C-C bond. The results are significant and deserve to be shown, because it can give more evidence to make sure whether there is a current flow along CBR’s bridge for big size starphene.

Response: The diagnosis of aromaticity in acenes by magnetic criteria has been controversial. For example, NICS(0) leads to highly shielded values, indicating a high degree of aromaticity in the innermost rings. Similarly, in the periphery C-C bonds, ring current strength integrations lead to large values. Table 1 collects the ring current strengths of the circulations in the different regions where these currents flow (plane A for the peripheral current). For example, the integration of pπJind across the C-C bond (of the inner 6-MR) of the [10]starphene arm leads to a value of 15.37 nA/T, which is almost the sum of the circulation of the local 6-MR, anthracene, and peripheral currents (16.06 nA/T). This has been clarified in the main text. However, it seems appropriate to discuss the strengths of each circulation to compare the degree of aromaticity among the same group of starphenes and with linear acenes. This led us to conclude that the aromaticity in these systems decreases with increasing size and is less aromatic than linear acenes. Considering the C-C bond strengths of the arm perimeter could lead to a misdiagnosis, such as considering these systems to be more aromatic than benzene itself. This is the opposite of what the values in Table 1 show.

Comment 4. If the isosurfaces maps of pπcan replaced or be added in the figures, they would be much more visual and comprehensible.

Response: Bindz isosurfaces are indeed quite useful to show the areas of shielding and deshielding, especially in non-planar molecules. Here we have taken advantage of the fact that these molecules are planar to screen different shielding and deshielding values using Bindz isosurfaces in and transverse to the molecular plane.  In this representation, it is also possible to appreciate the size of the cones. Therefore, we have decided to keep the Bindz isolines. 

Comment 5. There are several papers showing that starting from heptacene these species have a large diradical character and are better represented as a singlet open-shell species obtained using an unrestricted formalism. For this reason, although the systems in this paper are not linear acenes, they are non-linear acenes, they may have the similar properties. I kindly ask the authors: To acknowledge that the ground state of acenes is a controversial issue.

Comment 6. To cite references of articles showing that acenes have diradical character already starting from n = 7 and that long acenes have polyradical character ( see for instance, M. Bendikov et al. J. Am. Chem. Soc. 126 (2004) 7416; D. Jiang & S. Dai, J. Phys. Chm. A 112 (2008) 332; J. Hachmann et al. J. Chem. Phys. 127 (2007) 134309; J. B. Schriber et al. J. Chem. Theory Comput. 14 (2018) 6295)

Comment 7. To warn the reader that the conclusions one obtains on aromaticity trends considering long acenes as closed-shell species is quite different from the one obtained performing open-shell calculations (see J. Poater et al. J. Phys. Chem. A. 109 (2005) 10629)

Responses: We thank the reviewer for making us aware of the possible consequences of using larger arms in starphenes. We have added the suggested references in the text, as well as a warning:

"Since the arms are prone to reproduce the magnetic response of linear acenes, open-shell calculations should be considered. It is well-known that from [6]acene onwards, the determination of the ground state is controversial, as a radical character exists.[45-47] Reports of open-shell NICS(0) calculations on long [n]acenes (n > 7) indicate an antiaromatic character,[46] which could make the synthesis of starphenes with larger acene-like arms even more difficult."

Finally, we thank the reviewer for his valuable comments that have helped to improve the revised version of our manuscript.